# Ischemic Stroke in a Patient with Stable CADASIL during COVID-19: A Case Report

**DOI:** 10.3390/brainsci11121615

**Published:** 2021-12-08

**Authors:** Alessandro Cruciani, Fabio Pilato, Mariagrazia Rossi, Francesco Motolese, Vincenzo Di Lazzaro, Fioravante Capone

**Affiliations:** Neurology, Neurophysiology and Neurobiology Unit, Department of Medicine, Università Campus Bio-Medico di Roma, 00128 Rome, Italy; f.pilato@unicampus.it (F.P.); m.rossi@unicampus.it (M.R.); f.motolese@unicampus.it (F.M.); v.dilazzaro@unicampus.it (V.D.L.); f.capone@unicampus.it (F.C.)

**Keywords:** cerebral autosomal dominant arteriopathy with subcortical infarcts and leukoencephalopathy (CADASIL), stroke, hypercoagulability, COVID-19, SARS-CoV-2, endothelial dysfunction

## Abstract

Background: SARS-CoV-2 infection has been associated with different neurological conditions such as Guillain-Barré, encephalitis and stroke. Cerebral autosomal dominant arteriopathy with subcortical infarcts and leukoencephalopathy (CADASIL) is an inherited small-vessel disease characterized by recurrent ischemic stroke, cognitive decline, migraine and mood disturbances. One of the mechanisms involved in CADASIL pathogenesis is endothelial dysfunction, which causes an increased risk of recurrent strokes. Since COVID-19 infection is also associated with coagulopathy and endothelial dysfunction, the risk of ischemic stroke might be even higher in this population. We describe the case of a CADASIL patient who developed an acute ischemic stroke after SARS-CoV-2 infection. In patients with diseases causing endothelial dysregulation, such as CADASIL, the hypercoagulability related to COVID-19 may contribute to the risk of stroke recurrence.

## 1. Introduction

SARS-CoV-2 can impact the nervous system, and different neurological conditions, such as Guillain-Barré, encephalitis and cerebrovascular disease, have been associated with coronavirus disease (COVID-19) [1]. Herein, we report the case of a patient affected by cerebral autosomal dominant arteriopathy with subcortical infarcts and leukoencephalopathy (CADASIL) who developed an acute ischemic stroke after SARS-CoV-2 infection.

## 2. Case Report

A 60-year-old female patient with a medical history of hypertension came to our attention because of several neurological deficits that had developed over the last few years, significantly impairing her daily life. Four years earlier, she developed sudden weakness and hypoesthesia of the right hand. The symptoms resolved in a few days and no specific diagnostic tests were performed. Two months later, she developed hypoesthesia and weakness of the right lower limb. On neurological examination at the time, she had spastic gait, ataxia, slight pronation of the right upper limb and bilateral Babinski sign. Brain MRI showed extensive white matter hyperintensities (WMHs), so leukodystrophy was suspected. However, these WMHs were located bilaterally in the corona radiata, basal ganglia, the anterior part of the temporal lobes and the medium cerebellar peduncle (Figure 1A–D), and were highly suggestive of CADASIL. Genetic testing was performed, showing heterozygous mutation of the NOTCH3 gene (c.994 C<T; exon 6). The diagnosis of CADASIL was confirmed and antiplatelet prevention therapy was started. Since then, her clinical conditions remained stable, and the lesion load was unchanged at follow-up brain MRIs for 4 years until November 2020, when the patient was diagnosed with COVID-19 after a PCR nasal swab. The patient developed only mild respiratory symptoms, not requiring hospitalization or any specific treatment. Fifteen days after the COVID-19 diagnosis, she suddenly developed aphasia, agraphia and worsened right upper limb motor deficit, but she did not seek medical attention. Some days later, she reported these symptoms to her family medical doctor, and a new brain MRI was performed, showing a subacute ischemic area in the left corona radiata (Figure 1E,F). Therapy with acetylsalicylic acid was switched to clopidogrel as secondary prevention, while her symptoms improved in the next few weeks. The patient underwent a carotid doppler ultrasound and an echocardiogram, which did not reveal any pathological changes. The review of the blood pressure log, both in-hospital and the personal one the patient had kept, excluded uncontrolled hypertension.

## 3. Discussion

CADASIL is an autosomal dominant inherited disease caused by pathogenic mutations involving the NOTCH3 gene [2], which encodes a large transmembrane receptor, mainly expressed on vascular smooth muscle cells of the blood vessels and pericytes. The presence of this mutated receptor would lead to a dysregulation of the endothelial cells, which become more sensitive to oxidative stress, hypoxia and mechanical injuries, eventually leading to vessel occlusion and stroke [3]. The clinical manifestations of the disease are heterogeneous but most commonly include migraine with aura, mood disturbances, cognitive impairment evolving into early-onset dementia, and recurrent strokes [2,4]. The presence of white matter hyperintensities upon T2-weighted imaging and fluid-attenuated inversion recovery (FLAIR) at MRI is crucial for diagnosis, especially when located in the external capsule, the anterior part of the temporal lobes, the basal ganglia and the thalamus [2]. COVID-19 causes dysregulation of the coagulation system [5] even if the pathogenic mechanisms are still far from being fully known. Direct virus-mediated endothelial damage has been proposed, since SARS-CoV-2 can penetrate the endothelial cells via the ACE2 receptor, causing endothelial injury [5,6]. However, this mechanism alone could not explain the large number of thrombotic events in these patients; therefore, additional mechanisms related to the activation of the tissue factor and complement system [5] have been proposed.

## 4. Conclusions

Ours is the second case report of a CADASIL patient who experienced an ischemic stroke during COVID-19 infection [7]. We hypothesize that in patients with disorders caused by dysregulation of endothelial cells, hypercoagulability due to COVID-19 might increase the risk of cerebrovascular events. Indeed, a similar mechanism has been proposed in the pathogenesis of COVID-19-related posterior reversible encephalopathy syndrome (PRES), another neurological condition in which endothelial dysfunction and coagulopathy, together with a cytokine storm due to the infection, may lead to pathological changes [8].

Accordingly, in patients with genetic conditions characterized by stroke recurrence, such as CADASIL, we should be aware of an even higher risk of ischemic stroke in the case of SARS-CoV-2 infection. Careful evaluation and active monitoring of these vulnerable patients in the time of pandemics is thus mandatory.

## Figures and Tables

**Figure 1 brainsci-11-01615-f001:**
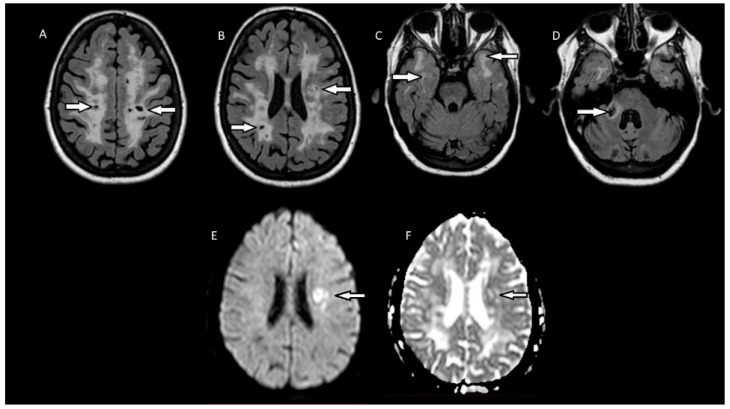
(**A**–**D**): MRI fluid-attenuated inversion recovery (FLAIR) scans showing multiple hypointense lesions (dilated perivascular spaces) and diffuse confluent white-matter hyperintensities in the corona radiata, basal ganglia, the anterior part of the temporal lobes and the medium cerebellar peduncle. (**E**,**F**): Diffusion-weighted MRI (DWI) and the apparent diffusion coefficient (ADC) revealed a restricted diffusion area on the left corona radiata, showing an acute ischemic lesion.

## Data Availability

Data are available upon request.

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
