# Peer review of "Ischemic Stroke in a Patient with Stable CADASIL during COVID-19: A Case Report"

_brainsci, 2021, doi:10.3390/brainsci11121615_

Round 1

Reviewer 1 Report

The article presented for review is a case report describing a 60-year-old patient with CADASIL complicated by an ischemic stroke caused by COVID-19.

In its present form, it isn’t easy to consider the manuscript ready for publication. This is a description of CADASIL, diagnosed four years before the coronavirus infection. The study lacks primary patient data, clinical data on stroke, and an assessment of the relationship between CADASIL, stroke, and COVID-19. The authors did not substantiate whether CADASIL is a risk factor for ischemic stroke in COVID-19. There are no grounds for the authors' conclusions - did the patient have basic laboratory parameters assessing platelet and endothelial function and hemostasis parameters performed? How was the patient treated? Besides, a similar case has already been reported.

Author Response

We thank the editor and the reviewer for the careful and insightful review of our manuscript. Below we answered to the points raised.

  • Provide primary patient data, clinical data on stroke: Thanks for the suggestion. We now provided some more details about patient’s clinical history, integrating what we had already included. It should be clearer now the hiatus between the pre- and post- SARS-CoV2 infection phase of patient’s story.   

  • Explain the assessment of the relationship between CADASIL, stroke, and COVID-19: Again, thanks for the suggestion. We acknowledge that this part might sound a little bit confusing. For these reasons, we have added some lines about the diagnostic workup – we did not include any info in the first place because everything was normal – and some other points in the discussion about COVID-19 and cerebrovascular disease.

  • Showing the basic laboratory parameters assessing platelet and endothelial function and haemostasis parameters: Unfortunately, the patient did not seek medical attention after symptoms onset. The MRI scan was performed at an Outpatient imaging center. However, she had some blood tests some days after the SARS-CoV2 infection – so they were pretty recent at the time of symptom onset – not showing anything abnormal, including hemoglobin, platelet count and coagulation parameters.  

  • Explain how was the patient treated: She did not receive any revascularization treatment because of timing (she reported symptoms days after their onset). Even if not part of clinical guidelines, we decided to start clopidogrel and stop therapy with salicylic acid.

Reviewer 2 Report

This is a single case report of a patient with known CADASIL who developed lacunar stroke 2 weeks after minorly-symptomatic COVID-19 infection. The case report is well-written (except for numerous grammatical errors). However, we know that COVID causes an increased risk of stroke so it isn’t a surprise that patients with a genetic risk of stroke (from CADASIL) would have stroke while infected with COVID. If this is published, the conclusion that “we should consider symptoms related to cerebrovascular event as an atypical event of COVID-19 infection” has a few problems.

-First, it is not atypical for COVID-19 patients to have stroke.

-Second, “cerebrovascular event” is non-specific and includes ICH/SAH/ischemic stroke /TIA. Make this more specific to ischemic stroke.

-Last, and most important, you are reporting on one patient – this report should not change practice. I suggest something like “symptoms of ischemic stroke could possibly be related to COVID-19 infection” instead.

- Also there numerous grammatical errors – too many to point out individually. Suggest having a native English speaker review this.

Author Response

We thank the editor and the reviewer for the careful and insightful review of our manuscript. Please see the attachment for the answer to the points raised.

Reviewer 3 Report

In this paper the authors describe a case of acute ischemic stroke in a patient with CADASIL who had COVID-19 infection with the hypothesis that COVID-19 related endothelial dysfunction and hypercoagulable state can lead to increased risk of stroke in these patients.

The case report is interesting especially in the era of COVID-19 pandemic. 

The research report is well structured, well-written, with clear introduction, case presentation and discussion/conclusion. The presentation is pretty clear, the references are enough and adequate. MR imaging helps illustrate the findings. The figures support the case report and are of adequate quality.

  • Would recommend to change RMI to MRI and WHM to WMH in the manuscript 

Author Response

We thank the editor and the reviewer for the careful and insightful review of our manuscript. Below we answered to the point raised.

  • We changed "RMI to MRI" and "WHM to WMH". 

Round 2

Reviewer 1 Report

The authors significantly extended the manuscript content with the necessary elements. The paper is now ready to be accepted.